# Minimal Variance Sampling in Stochastic Gradient Boosting

**Bulat Ibragimov**
Yandex, Moscow, Russia
Moscow Institute of Physics and Technology
`ibrbulat@yandex.ru`

**Gleb Gusev**
Sberbank*, Moscow, Russia
`gusev.g.g@sberbank.ru`

## Abstract

Stochastic Gradient Boosting (SGB) is a widely used approach to regularization of boosting models based on decision trees. It was shown that, in many cases, random sampling at each iteration can lead to better generalization performance of the model and can also decrease the learning time. Different sampling approaches were proposed, where probabilities are not uniform, and it is not currently clear which approach is the most effective. In this paper, we formulate the problem of randomization in SGB in terms of optimization of sampling probabilities to maximize the estimation accuracy of split scoring used to train decision trees. This optimization problem has a closed-form nearly optimal solution, and it leads to a new sampling technique, which we call *Minimal Variance Sampling (MVS)*. The method both decreases the number of examples needed for each iteration of boosting and increases the quality of the model significantly as compared to the state-of-the art sampling methods. The superiority of the algorithm was confirmed by introducing MVS as a new default option for subsampling in CatBoost, a gradient boosting library achieving state-of-the-art quality on various machine learning tasks.

## 1 Introduction

Gradient boosted decision trees (GBDT) [16] is one of the most popular machine learning algorithms as it provides high-quality models in a large number of machine learning problems containing heterogeneous features, noisy data, and complex dependencies [31]. There are many fields where gradient boosting achieves state-of-the-art results, e.g., search engines [34, 3], recommendation systems [30], and other applications [36, 4].

One problem of GBDT is the computational cost of the learning process. GBDT may be described as an iterative process of constructing decision tree models, each of which estimates negative gradients of examples' errors. At each step, GBDT greedily builds a tree. GBDT scores every possible feature split and chooses the best one, which requires computational time proportional to the number of data instances. Since most GBDT models consist of an ensemble of many trees, as the number of examples grows, more learning time is required, what imposes restrictions on using GBDT models for large industry datasets.

Another problem is the trade-off between the capacity of the GBDT model and its generalization ability. One of the most critical parameters that influence the capacity of boosting is the *number of iterations* or the size of the ensemble. The more components are used in the algorithm, the more complex dependencies can be modeled. However, an increase in the number of models in the ensemble does not always lead to an increase in accuracy and, moreover, can decrease its generalization

ability [28]. Therefore boosting algorithms are usually provided with regularization methods, which are needed to prevent overfitting.

A common approach to handle both of the problems described above is to use random subsampling [17] (or bootstrap) at every iteration of the algorithm. Before fitting a next tree, we select a subset of training objects, which is smaller than the original training dataset, and perform learning algorithm on a chosen subsample. The fraction of chosen objects is called *sample rate*. Studies in this area show that this demonstrates excellent performance in terms of learning time and quality [17]. It helps to speed up the learning process for each decision tree model as it uses less data. Also, the accuracy can increase because, despite the fact that the variance of each component of the ensemble goes up, the pairwise correlation between trees of the ensemble decreases, what can lead to a reduction in the total variance of the model.

In this paper, we propose a new approach to theoretical analysis of random sampling in GBDT. In GBDT, random subsamples are used for evaluation of candidate splits, when each next decision tree is constructed. Random sampling decreases the size of the active training dataset, and the training procedure becomes more noisy, what can entail a decrease in the quality. Therefore, sampling algorithm should select the most informative training examples, given a constrained number of instances the algorithm is allowed to choose. We propose a mathematical formulation of this optimization problem in SGB, where the accuracy of estimated scores of candidate splits is maximized. For every fixed sample rate (ratio of sampled objects), we propose a solution to this *sampling problem* and provide a novel algorithm Minimal Variance Sampling (MVS). MVS relies on the distribution of loss derivatives and assigns probabilities and weights with which the sampling should be done. That makes the procedure adaptive to any data distribution and allows to significantly outperform the state of the art SGB methods, operating with way less number of data instances.

## 2 Background

In this section, we introduce necessary definitions and notation. We start from the GBDT algorithm, and then we describe its two popular modifications that use data subsampling: Stochastic Gradient Boosting [17] and Gradient-Based One-Side Sampling (GOSS) [24].

### 2.1 Gradient Boosting

Consider a dataset $\{\vec{x_i}, y_i\}_{i=1}^N$ sampled from some unknown distribution $p(\vec{x}, y)$. Here $\vec{x_i} \in \mathcal{X}$ is a vector from the $d$-dimensional vector space. Value $y_i \in \mathbb{R}$ is the response to the input $\vec{x_i}$ (or *target*). Given a loss function $L : \mathbb{R}^2 \to \mathbb{R}_+$, the problem of supervised learning task is to find function $F : X \to \mathbb{R}$, which minimizes the empirical risk:

$$\hat{\mathcal{L}}(F) = \sum_{i=1}^N L(F(\vec{x_i}), y_i)$$

*Gradient boosting (GB)* [16] is a method of constructing the desired function $F$ in the form

$$F(\vec{x}) = \sum_{k=1}^n \alpha f_k(\vec{x}),$$

where $n$ is *the number of iterations*, i.e., the amount of base functions $f_k$ chosen from a simple parametric family $\mathcal{F}$, such as linear models or decision trees with small depth. The *learning rate*, or step size in functional space, is denoted by $\alpha$. Base learners $\{f_k\}$ are learned sequentially in the following way. Given a function $F_{m-1} = \sum_{i=1}^{m-1} \alpha f_k$, the goal is to construct the next member $f_m$ of the sequence $f_1, \ldots, f_{m-1}$ such that:

$$f_m = \arg\min_{f \in \mathcal{F}} \hat{\mathcal{L}}(F_{m-1} + f) = \arg\min_{f \in \mathcal{F}} \sum_{i=1}^N L(F_{m-1}(\vec{x_i}) + f(\vec{x_i}), y_i) \tag{1}$$

Gradient boosting constructs a solution of Equation 1 by calculating first-order derivatives (gradients) $g_i^m(\vec{x_i}, y_i) = \left. \frac{\partial L(\hat{y_i}, y_i)}{\partial \hat{y_i}} \right|_{\hat{y_i} = F_{m-1}(\vec{x_i})}$ of $\hat{\mathcal{L}}(F)$ at point $F_{m-1}$ and performing a negative gradi-

ent step in the functional space of examples $\{\vec{x_i}, y_i\}_{i=1}^N$. The latter means that $f_m$ is learned using $\{-g_i^m(\vec{x_i}, y_i)\}_{i=1}^N$ as targets and is fitted by the least–squares approximation:

$$f_m = \arg\min_{f \in \mathcal{F}} \sum_{i=1}^N (f(\vec{x_i}) - (-g_i^m(\vec{x_i}, y_i)))^2 \qquad (2)$$

If a subfamily of decision tree functions is taken as a set of base functions $\mathcal{F}$ (e.g., all decision trees of depth 5), the algorithm is called *Gradient Boosted Decision Trees (GBDT)* [16]. A decision tree divides the original feature space $\mathbb{R}^d$ into disjoint areas, also called *leaves*, with a constant value in each region. In other words, the result of a decision tree learning is a disjoint union of subsets $\{X_1, X_2, ..., X_q : \bigsqcup_{i=1}^q X_i = \mathcal{X}\}$ and a piecewise constant function $f(\vec{x}) = \sum_{i=1}^q \mathbb{I}\{\vec{x} \in X_i\} c_i$. The learning procedure is recursive. It starts from the whole set $\mathbb{R}^d$ as the only region. For every of the already built regions, the algorithm looks out all split candidates by one feature and sets a *score* for each split. The score is usually a measure of accuracy gain based on target distributions in the regions before and after splitting. The process continues until a stopping criterion is reached.

Besides the classical gradient descent approach to GBDT defined by Equation 2, we also consider a second-order method based on calculating diagonal elements of hessian of empirical risk

$$h_i^m(\vec{x_i}, y_i) = \frac{\partial^2 L(\hat{y}_i, y_i)}{\partial \hat{y}_i^2}\bigg|_{\hat{y}_i = F_{m-1}(\vec{x_i})}.$$

The rule for choosing the next base function in this method [8] is:

$$f_m = \arg\min_{f \in \mathcal{F}} \sum_{i=1}^N h_i^m(\vec{x_i}, y_i) \left( f(\vec{x_i}) - \left( -\frac{g_i^m(\vec{x_i}, y_i)}{h_i^m(\vec{x_i}, y_i)} \right) \right)^2 \qquad (3)$$

## 2.2 Stochastic Gradient Boosting

Stochastic Gradient Boosting is a randomized version of standard Gradient Boosting algorithm. Motivated by Breiman's work about adaptive bagging [2], Friedman [17] came to the idea of adding randomness into the tree building procedure by using a subsampling of the full dataset. For each iteration of the boosting process, the sampling algorithm of SGB selects random $s \cdot N$ objects without replacement and *uniformly*. It effectively reduces the complexity of each iteration down to the factor of $s$. It is also proved by experiments [17] that, in some cases, the quality of the learned model can be improved by using SGB.

## 2.3 GOSS

SGB algorithm makes all objects to be selected equally likely. However, different objects have different impacts on the learning process. Gradient-based one-side sampling (GOSS) implements an idea that objects $\vec{x_i}$ with larger absolute value of the gradient $|g_i|$ are more important than the ones that have smaller gradients. A large gradient value indicates that the model can be improved significantly with respect to the object, and it should be sampled with higher probability compared to well-trained instances with small gradients. So, GOSS takes the most important objects with probability 1 and chooses a random sample of other objects. To avoid distribution bias, GOSS re-weighs selected samples by setting higher weights to the examples with smaller gradients. More formally, the training sample consists of $top\_rate \times N$ instances with largest $|g_i|$ with weight equal to 1 and of $other\_rate \times N$ instances from the rest of the data with weights equal to $\frac{1-top\_rate}{other\_rate}$.

## 3 Related work

A common approach to randomize the learning of GBDT model is to use some kind of SGB, where instances are sampled equally likely or *uniformly*. This idea was implemented in different ways. Originally, Friedman proposed to sample a subset of objects of a fixed size [17] without replacement. However, in today's practice, other similar techniques are applied, where the size of the subset can be stochastic. For example, the objects can be sampled independently using a Bernoulli process [24],

or a bootstrap procedure can be applied [14]. To the best of our knowledge, GOSS proposed in [24] is the only weighted (non-uniform) sampling approach applied to GBDT. It is based on intuitive ideas, but its choice is empirical. Therefore our theoretically grounded method MVS outperforms GOSS in experiments.

Although, there is a surprising lack of non-uniform sampling for GBDT, there are [13] adaptive weighted approaches proposed for AdaBoost, another popular boosting algorithm. These methods mostly rely on weights of instances defined in the loss function at each iteration of boosting [15, 35, 9, 26, 20]. These papers are mostly focused on the accurate estimation of the loss function, while subsamples in GBDT are used to estimate the scores of candidate splits, and therefore, sampling methods of both our paper and GOSS are based on the values of gradients. GBDT algorithms do not apply adaptive weighting of training instances, and methods proposed for AdaBoost cannot be directly applied to GBDT.

One of the most popular sampling methods based on target distribution is Importance Sampling [37] widely used in deep learning [21]. The idea is to choose the objects with larger loss gradients with higher probability than with smaller ones. This leads to a variance reduction of mini-batch estimated gradient and has a positive effect on model performance. Unfortunately, Importance Sampling poorly performs for the task of building decision trees in GBDT, because the score of a split is a ratio function, which depends on the sum of gradients and the sample sizes in leaves, and the variance of their estimations all affect the accuracy of the GBDT algorithm. The following part of this paper is devoted to a theoretically grounded method, which overcomes these limitations.

# 4 Minimal Variance Sampling

## 4.1 Problem setting

As it was mentioned in Section 2.1, training a decision tree is a recursive process of selecting the best data partition (or *split*), which is based on a value of some feature. So, given a subset $A$ of original feature space $\mathcal{X}$, split is a pair of feature $f$ and its value $v$ such that data is partitioned into two sets: $A_1 = \{\vec{x} \in A : x_f < v\}$, $A_2 = \{\vec{x} \in A : x_f \geq v\}$. Every split is evaluated by some *score*, which is used to select the best one among them.

There are various scoring metrics, e.g., Gini index and entropy criterion [32] for classification tasks, mean squared error (MSE) and mean absolute error (MAE) for regression trees. Most of GB implementations (e.g. [8]) consider hessian while learning next tree (second-order approximation). The solution to in a leaf $l$ is the constant equal to the ratio $\frac{\sum_{i \in l} g_i}{\sum_{i \in l} h_i}$ of the sum of gradients and the sum of hessian diagonal elements. The score $S(f, v)$ of a split $(f, v)$ is calculated as

$$S(f, v) := \sum_{l \in L} \frac{\left( \sum_{i \in l} g_i \right)^2}{\sum_{i \in l} h_i}, \tag{4}$$

where $L$ is the set of obtained leaves, and leaf $l$ consists of objects that belong to this leaf. This score is, up to a common constant, the opposite to the value of the functional minimized in Equation 3 when we add this split to the tree. For classical GB based on the first-order gradient steps, according to Equation 2, score $S(f, v)$ should be calculated by setting $h_i = 1$ in Equation 4.

To formulate the problem, we first describe the general sampling procedure, which generalizes SGB and GOSS. Sampling from a set of size $N$ may be described as a sequence of random variables $(\xi_1, \xi_2, ..., \xi_N)$, where $\xi_i \sim \text{Bernoulli}(p_i)$, and $\xi_i = 1$ indicates that $i$-th example was sampled and should be used to estimate scores $S(f, v)$ of different candidates $(f, v)$. Let $n_{sampled}$ be the number of selected instances. By *sampling with sampling ratio $s$*, we denote any sequence $(\xi_1, \xi_2, ..., \xi_N)$, which samples $s \times 100\%$ of data on average:

$$\mathbb{E}(n_{sampled}) = \mathbb{E} \sum_{i=1}^{N} \xi_i = \sum_{i=1}^{N} p_i = N \cdot s. \tag{5}$$

To make all key statistics (sum of gradients and sum of hessians in the leaf) unbiased, we perform inverse probability weighting estimation (IPWE) [18], which assigns weight $w_i = \frac{1}{p_i}$ to instance $i$.

In GB with sampling, score $S(f, v)$ is approximated by

$$\hat{S}(f, v) := \sum_{l \in L} \frac{\left( \sum_{i \in l} \frac{1}{p_i} \xi_i g_i \right)^2}{\sum_{i \in l} \frac{1}{p_i} \xi_i h_i}, \tag{6}$$

where the numerator and denominator are estimators of $\left( \sum_{i \in l} g_i \right)^2$ and $\sum_{i \in l} h_i$ correspondingly.

We are aimed at minimization of squared deviation $\Delta^2 = \left( \hat{S}(f, v) - S(f, v) \right)^2$. Deviation $\Delta$ is a random variable due to the randomness of the sampling procedure (randomness of $\xi_i$). Therefore, we consider the minimization of the expectation $\mathbb{E}\Delta^2$.

**Theorem 1.** *The expected squared deviation $\mathbb{E}\Delta^2$ can be approximated by*

$$\mathbb{E}\Delta^2 \approx \sum_{l \in L} c_l^2 \left( 4Var(x_l) - 4c_l Cov(x_l, y_l) + c_l^2 Var(y_l) \right), \tag{7}$$

*where $x_l := \sum_{i \in l} \frac{1}{p_i} \xi_i g_i$, $y_l := \sum_{i \in l} \frac{1}{p_i} \xi_i h_i$, and $c_l := \frac{\mu_{x_l}}{\mu_{y_l}} = \frac{\sum_{i \in l} g_i}{\sum_{i \in l} h_i}$ is the value in the leaf $l$ that would be assigned if $l$ would be a terminal node of the tree.*

The proof of this theorem is available in the Supplementary Materials.

The term $-4c_l Cov(x_l, y_l)$ in Equation 7 has an upper bound of $\left( 4Var(x_l) + c_l^2 Var(y_l) \right)$. Using Theorem 1, we come to an upper bound minimization problem

$$\sum_{l \in L} c_l^2 \left( 4Var(x_l) + c_l^2 Var(y_l) \right) \to \min. \tag{8}$$

Note that we do not have the values of $c_l$ for all possible leaves of all possible candidate splits in advance, when we perform sampling procedure. A possible approach to Problem 8 is to substitute all $c_l^2$ by a universal constant value, which is a parameter of sampling algorithm. Also, note that $Var(x_l)$ is $\sum_{i \in l} \frac{1}{p_i} g_i^2$ and $Var(y_l)$ is $\sum_{i \in l} \frac{1}{p_i} h_i^2$ up to constants that do not depend on the sampling procedure. In this way, we come to the following form of Problem 8:

$$\sum_{i=1}^{N} \frac{1}{p_i} g_i^2 + \lambda \sum_{i=1}^{N} \frac{1}{p_i} h_i^2 \to \min_{p_i}, \quad \text{w.r.t.} \quad \sum_{i=1}^{N} p_i = N \cdot s \quad \text{and} \quad \forall i \in \{1, \dots, N\} \; p_i \in [0, 1]. \tag{9}$$

### 4.2  Theoretical analysis

Here we show that Problem 9 has a simple solution and leads to an effective sampling algorithm. First, we discuss its meaning in the case of first-order optimization, where we have $h_i = 1$.

The first term of the minimized expression is responsible for gradient distribution over the leaves of the decision tree, while the second one is responsible for the distribution of sample sizes. Coefficient $\lambda$ controls the magnitude of each of the component. It can be seen as a tradeoff between the variance of a single model and the variance of the ensemble. The variance of the ensemble consists of individual variances of every single algorithm and pairwise correlations between models. On the one hand, it is crucial to reduce individual variances of each model; on the other hand, the more dissimilar subsamples are, the less the total variance of the ensemble is. This is reflected in the accuracy dependence on the number of sampled examples: the slight reduction of this number usually leads to increase in the quality as the variance of each model is not corrupted a lot, but, when the sample size goes down to smaller numbers, the sum of variances prevails over the loss in correlations and the accuracy dramatically decreases.

It is easy to derive that setting $\lambda$ to 0 implies the procedure of Importance Sampling. As it was mentioned before, the applicability of this procedure in GBDT is constrained since it is still important to

estimate the number of instances accurately in each node of the tree. Besides, Importance Sampling is suffering from numerical instability while dealing with small gradients close to zero, what usually happens on the latter gradient boosting iterations. In this case, the second part of the expression may be interpreted as a regularisation member prohibiting enormous weights.

Setting $\lambda$ to $\infty$ implies the SGB algorithm.

For arbitrary $\lambda$ general solution is given by the following theorem (we leave the proof to the Supplementary Materials):

**Theorem 2.** *There exists a value $\mu$ such that $p_i = \min\left(1, \frac{\sqrt{g_i^2 + \lambda h_i^2}}{\mu}\right)$ is a solution to Problem 9.*

For abbreviations, everywhere below, we refer to the expression $\hat{g}_i = \sqrt{g_i^2 + \lambda h_i^2}$ using *regularized absolute value* term. The number $\mu$ defined above is a threshold for decision, whether to pick an example deterministic of by coin flipping. From the solution we see, that for any data instance, the weight is always bounded by some number, so the estimator is more computationally stable than IPWE usually is.

From Theorem 2, we conclude that optimal sampling scheme in terms of Equation 9 is described by Algorithm 1:

---
**Algorithm 1** MVS Algorithm

---
  **Input:** $X$, $y$, $Loss$, $maxIter$, $sampleRate$, $\lambda$
  $ensemble = []$
  $ensemble$.**append**(**InitialGuess**$(X, y)$)
  **for** $i$ **from** $1$ **to** $maxIter$ **do**
    $predictions = ensemble$.**predict**$(X)$
    $gradients$, $hessians$ = **CalculateDerivatives**$(Loss, y, predictions)$
    $regGradients[i] = \sqrt{gradients[i]^2 + \lambda \cdot hessians[i]^2}$
    $\mu$ = **CalculateThreshold**$(regGradients, sampleRate)$
    $probs[i]$ = **Min**$(\frac{regGradients[i]}{\mu}, 1)$
    $weights[i] = \frac{1}{probs[i]}$
    $idxs$ = **Select**$(probs)$
    $nextTree$ = **TrainTree**$(X[idxs], -gradients[idxs], hessians[idxs], weights[idxs])$
    $ensemble$.**append**$(nextTree)$
  **end for**

---

### 4.3 Algorithm

Now we are ready to derive the MVS algorithm from Theorem 2, which can be directly applied to general scheme of Stochastic Gradient Boosting. First, for given sample rate $s$, MVS finds the threshold $\mu$ to decide, which gradients are considered to be large. Example $i$ with regularized absolute value $\sqrt{g_i^2 + \lambda h_i^2}$ higher than chosen $\mu$ is sampled with probability equal to 1. Every object with small gradient is sampled independently with probability $p_i = \frac{\sqrt{g_i^2 + \lambda h_i^2}}{\mu}$ and is assigned weight $w_i = \frac{1}{p_i}$. Still, it is not apparent how to find such a threshold $\mu^*$ that will give the required sampling ratio $s = s^*$.

A brute-force algorithm relies on the fact that the sampling ratio has an inverse dependence on the threshold: the higher the threshold, the lower the fraction of sampled instances. First, we sort the data by regularized absolute value in descending order. Note that now, given a threshold $\mu$, the sampling ratio $s$ can be calculated as $s(\mu) = \frac{1}{\mu} \sum_{i=k+1}^{N} \sqrt{g_i^2 + \lambda h_i^2} + k$, where $k+1$ is the number of the first element in sorted sequence, which is less than $\mu$. Then the binary search is applied to find a threshold $\mu^*$ with the desired property $s(\mu^*) = s^*$. To speed up this algorithm, the precalculation of cumulative sums of regularized absolute values $\sum_{i=k+1}^{N} \sqrt{g_i^2 + \lambda h_i^2}$ for every $k$ is performed, so the calculation of sampling ratio at each step of binary search has $O(1)$ time complexity. The total

complexity of this procedure is $O(N \log N)$, due to sorting at the beginning. To compare with, SGB and GOSS algorithms have $O(N)$ complexity for sampling.

We propose a more efficient algorithm, which is similar to the quick select algorithm [27]. In the beginning, the algorithm randomly selects the gradient, which is a candidate to be a threshold. The data is partitioned in such a way that all the instances with smaller gradients and larger gradients are on the opposite sides of the candidate. To calculate the current sample rate, it is sufficient to calculate the number of examples on the larger side and the sum of regularized absolute values on the other side. Then, estimated sample rate is used to determine the side where to continue the search for the desired threshold. If the current sample rate is higher, then algorithms searches threshold on the side with smaller gradients, otherwise on the side with greater. Calculated statistics for each side may be reused in further steps of the algorithm, so the number of the operations at each step is reduced by the number of rejected examples. The time complexity analysis can be carried out by analogy with the quick select algorithm, which results in $O(N)$ complexity.

---

**Algorithm 2** Calculate Threshold

---

Init $sumSmall = nLarge = 0$, $candidatesArray = all$
$length = \textbf{Length}(candidatesArray)$
$candidateThreshold = \textbf{RandomSelect}(candidatesArray)$
$mid = \textbf{Partition}(candidatesArray, candidate)$
$smallArray = candidatesArray[1,...,mid\text{-}1]$
$largeArray = candidatesArray[mid\text{+}1,...,length]$
$curSampleRate = \frac{\textbf{Sum}(smallArray)+sumSmall}{candidateThreshold} + \textbf{Length}(largeArray) + nLarge + 1$
**if Length**$(smallArray) == 0$ **and** $curSampleRate < sampleRate$ **then**
    **return** $\frac{sumSmall}{sampleRate-nLarge-\textbf{Length}(largeArray)-1}$
**else if Length**$(largeArray) == 0$ **and** $curSampleRate > sampleRate$ **then**
    **return** $\frac{sumSmall+\textbf{Sum}(smallArray)+candidateThreshold}{sampleRate-nLarge}$
**else if** $CurSampleRate > sampleRate$ **then**
    $sumSmall = sumSmall + \textbf{Sum}(smallArray) + candidateThreshold$
    **return CalculateThreshold**$(largeArray, sumSmall, nLarge, sampleRate)$
**else**
    $nLarge = nLarge + \textbf{Length}(largeArray) + 1$
    **return CalculateThreshold**$(smallArray, sumSmall, nLarge, sampleRate)$
**end if**

---

# 5   Experiments

Here we provide experimental results of MVS algorithm on two popular open-source implementations of gradient boosting: CatBoost and LightGBM.

**CatBoost.** The default setting of CatBoost is known to achieve state-of-the-art quality on various machine learning tasks [29]. We implemented MVS in CatBoost and performed benchmark comparison of MVS with sampling ratio 80% and default CatBoost with no sampling on 153 publicly available and proprietary binary classification datasets of different sizes up to 45 millions instances. The algorithms were compared by the ROC-AUC metric, and we calculated the number of wins for each algorithm. The results show significant improvement over the existing default: 97 wins of MVS versus 55 wins of default setting and $+0.12\%$ mean ROC-AUC improvement.

The source code of MVS is publicly available [6] and ready to be used as a default option of CatBoost algorithm. The latter means that MVS is already acknowledged as a new benchmark in SGB implementations.

**LightGBM.** To perform a fair comparison with previous sampling techniques (GOSS and SGB), MVS was also implemented in LightGBM, as it is a popular open-source library with GOSS inside. The MVS source code for LightGBM may be found at [19].

Datasets' descriptions used in this section are placed in Table 1. All the datasets are publicly available and were preprocessed according to [5].

| Dataset | # Examples | # Features |
|---|---|---|
| KDD Internet [1] | 10108 | 69 |
| Adult [25] | 48842 | 15 |
| Amazon [23] | 32769 | 10 |
| KDD Upselling [11] | 50000 | 231 |
| Kick prediction [22] | 72983 | 36 |
| KDD Churn [10] | 50000 | 231 |
| Click prediction [12] | 399482 | 12 |

Table 1: Datasets description

We used the tuned parameters and train-test splitting for each dataset from [5] as baselines, presetting the sampling ratio to 1. For tuning sampling parameters of each algorithm (sample rate and $\lambda$ coefficient for MVS, large gradients fraction and small gradients fraction for GOSS, sample rate for SGB), we use 5-fold cross-validation on train subset of the data. Then the tuned models are evaluated on test subsets (which is 20% of the original size of the data). Here we use the $1 - \text{ROC-AUC}$ score as an error measure (lower is better). To make the results more statistically significant, the evaluation part is run 10 times with different seeds. The final result is defined as the mean over these 10 runs.

Here we also introduce hyperparameter-free MVS algorithm modification. Since $\lambda$ (see Equation 9) is an approximation of squared mean leaf value upper bound, we replace it with a squared mean of the initial leaf. As it will be shown, it achieves near-optimal results and dramatically reduces time spent on parameter tuning. Since it sets $\lambda$ adaptively at each iteration, we refer to this method as *MVS Adaptive*.

**Quality comparison.** The first experiments are devoted to testing MVS as a regularization method. We state the following question: how much the quality changes when using different sampling techniques? To answer this question, we tuned the sampling parameters of algorithms to get the best quality. This quality scores compared to baselines quality are presented in Table 2. From this results, we can see that MVS demonstrates the best generalization ability among given sampling approaches. The best parameter $\lambda$ for MVS is about $10^{-1}$, it shows good performance on most of the datasets. For GOSS, the best ratio of large and small gradients varies a lot from the predominance of large to the predominance of small.

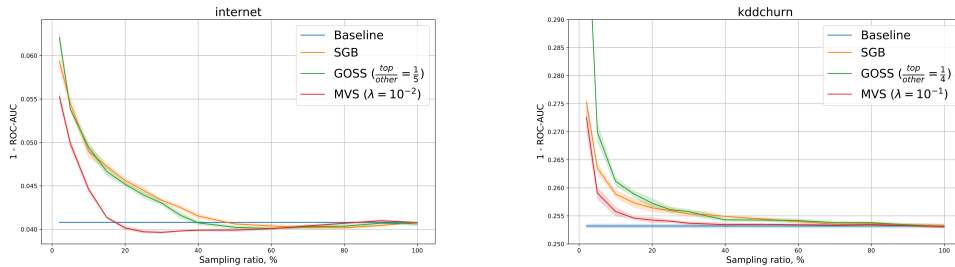

Figure 1: $1 - \text{ROC-AUC}$ error versus the fraction of sampled examples per one iteration

| | KDD Internet | Adult | Amazon | KDD Upselling | Kick | KDD Churn | Click | Average |
|---|---|---|---|---|---|---|---|---|
| Baseline | 0.0408 | 0.0688 | 0.1517 | 0.1345 | 0.2265 | **0.2532** | 0.2655 | -0.0% |
| SGB | -1.13% | +0.81% | -1.14% | +0.03% | -0.14% | +0.14% | **-0.14%** | -0.22% |
| GOSS | -0.64% | -0.11% | -1.23% | +0.07% | -0.10% | +0.16% | -0.09% | -0.28% |
| MVS | **-3.03%** | **-0.24%** | **-1.78%** | -0.07% | **-0.19%** | +0.17% | -0.04% | **-0.74%** |
| MVS Adaptive | -2.79% | -0.13% | -1.57% | **-0.28%** | **-0.19%** | +0.07% | -0.03% | -0.70% |

Table 2: Baseline scores / relative error change

| Sample rate | 0.02 | 0.05 | 0.1 | 0.15 | 0.2 | 0.25 | 0.3 | 0.35 | 0.4 | 0.5 |
|---|---|---|---|---|---|---|---|---|---|---|
| SGB | +19.92% | +11.35% | +6.83% | +4.99% | +3.84% | +3.03% | +2.17% | +1.57% | +1.10% | +0.42% |
| GOSS | +22.37% | +12.75% | +8.00% | +5.32% | +3.39% | +2.25% | +1.41% | +0.75% | +0.23% | -0.16% |
| MVS | +13.93% | +7.76% | **+3.69%** | +1.91% | +0.74% | +0.14% | **-0.21%** | **-0.43%** | **-0.41%** | -0.45% |
| MVS Adaptive | **+13.72%** | **+7.47%** | +3.71% | **+1.70%** | **+0.55%** | **-0.03%** | -0.07% | -0.28% | -0.32% | **-0.51%** |

Table 3: Relative error change, average over datasets

The next research question is whether MVS is capable of reducing sample size per each iteration needed to achieve acceptable quality. Furthermore, whether MVS is harmful to accuracy while using small subsamples. For this experiment, we tuned parameters, so that the algorithms achieve the baseline score (or their best score if it is not possible) using the least number of instances. Figure 1 shows the dependence of error on the sample size for two datasets and its $\pm\sigma$ confidence interval. Table 3 demonstrates average relative error change with respect to the baseline over all datasets used in this paper. From these results, we can conclude that MVS reaches the goal of reducing the variance of the models, and a decrease in sample size affects the accuracy much less than it does for other algorithms.

**Learning time comparison.** To compare the speed-up ability of MVS, GOSS and SGB, we used runs from the previous experiment setting, i.e., parameters were chosen in order to have the smallest sample rate with no quality loss. Among them, we choose the ones which have the least training time (if it is impossible to beat baseline, the best score point is chosen). The summary is shown in Table 4, which demonstrates the average learning time gain relative to the baseline learning time (using all examples). One can see that the usage of MVS has an advantage in training time over other methods at the amount of about 10% for datasets presented in this paper. Also, it is important to mention that tuning the hyperparemeters is a main part of training a model. There is one common hyperparameter for all sampling algorithms - sample rate. GOSS has one additional hyperparameter - ratio of large and small gradients in the subsample, and MVS has a hyperparameter $\lambda$. So tuning GOSS and MVS may potentially take more time than SGB. But introducing MVS Adaptive algorithm dramatically reduces tuning time due to hyperparameter-free sampling procedure, and we can conclude from Tables 2 and 3 that it achieves approximately optimal results on the test data

**Large datasets.** Experiments with CatBoost show that regularization effect of MVS is efficient for any size of the data. But for large datasets it is more crucial to reduce learning time of the model. To prove that MVS is efficient in accelerating the training we use Higgs dataset [33] (11000000 instances and 28 features) and Recsys datasets [7] (16549802 instances and 31 features). The set up of experiment remains the same as in the previous paragraph. For Higgs dataset SGB is not able to achieve the baseline quality with less than 100% sample size, while GOSS and MVS managed to do this with 80% of samples and MVS was faster than GOSS (-17.7% versus -8.5%) as it converges earlier. For Recsys dataset relative learning time differences are -50.3% for SGB (sample rate 20%), -39.9% for SGB (sample rate 20%) and -61.5% for MVS (sample rate 10%).

| | SGB | GOSS | MVS |
|---|---|---|---|
| time difference | -20.7% | -20.4% | -27.7% |

Table 4: Relative learning time change

# 6   Conclusion

In this paper, we addressed a surprisingly understudied problem of weighted sampling in GBDT. We proposed a novel technique, which directly maximizes the accuracy of split scoring, a core step of the tree construction procedure. We rigorously formulated this goal as an optimization problem and derived a near-optimal closed-form solution. This solution led to a novel sampling technique MVS. We provided our work with necessary theoretical statements and empirical observations that show the superiority of MVS over the well-known state-of-the-art approaches to data sampling in SGB. MVS is implemented and used by default in CatBoost open-source library. Also, one can find MVS implementation in LightGBM package, and its source code is publicly available for further research.

## Acknowledgements

We are deeply indebted to Liudmila Prokhorenkova for valuable contribution to the content and helpful advice about the presentation. We are also grateful to Aleksandr Vorobev for sharing ideas and support, Anna Veronika Dorogush and Nikita Dmitriev for experiment assistance.

## Footnotes

*The study was done while working at Yandex

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
