[Supplementary Material]

# MVS Supplementary Materials

**Bulat Ibragimov**
Yandex, Moscow, Russia
Moscow Institute of Physics and Technology
ibrbulat@yandex.ru

**Gleb Gusev**
Sberbank*, Moscow, Russia
gusev.g.g@sberbank.ru

## 1 Proof of Theorem 1

*Proof.* We estimate the expectation by representing $\hat{S}(f, v)$ as the function $F(x_1, y_1, \ldots, x_{|L|}, y_{|L|}) := \sum_{l=1}^{|L|} \frac{x_l^2}{y_l}$.

We use the first-order Taylor series expansion of $F$ at the point $(\mu_{x_1}, \mu_{y_1}, \ldots, \mu_{x_{|L|}}, \mu_{y_{|L|}})$, where $\mu_{x_l} = \mathbb{E}x_l = \sum_{i \in l} g_i$ and $\mu_{y_l} = \mathbb{E}y_l = \sum_{i \in l} h_i$.

Without loss of generality, we further provide calculations for the case $|L| = 1$.

We have $F(x_1, y_1) \approx F(\mu_{x_1}, \mu_{y_1}) + 2\frac{\mu_{x_1}}{\mu_{y_1}}(x_1 - \mu_{x_1}) - \frac{\mu_{x_1}^2}{\mu_{y_1}^2}(y_1 - \mu_{y_1})$, and, therefore, $\Delta = F(x_1, y_1) - F(\mu_{x_1}, \mu_{y_1}) \approx 2\frac{\mu_{x_1}}{\mu_{y_1}}(x_1 - \mu_{x_1}) - \frac{\mu_{x_1}^2}{\mu_{y_1}^2}(y_1 - \mu_{y_1})$.

Further, we have

$$\mathbb{E}\Delta^2 \approx \mathbb{E}(2\frac{\mu_{x_1}}{\mu_{y_1}}(x_1 - \mu_{x_1}) - \frac{\mu_{x_1}^2}{\mu_{y_1}^2}(y_1 - \mu_{y_1}))^2 = c_1^2(4Var(x_1) - 4c_1Cov(x_1, y_1) + c_1^2Var(y_1)).$$

$\square$

## 2 Proof of Theorem 2

*Proof.* Our goal is to find solution to the optimization problem:

$$\sum_{i=1}^{N} \frac{1}{p_i} g_i^2 + \lambda \sum_{i=1}^{N} \frac{1}{p_i} h_i^2 \to \min_{p_i}, \quad \text{w.r.t.} \quad \sum_{i=1}^{N} p_i = N \cdot s \quad \text{and} \quad \forall i \, p_i \in [0, 1]. \tag{1}$$

Lagrange function for this problem has form:

$$\mathcal{L} = \sum_{i=1}^{N} \frac{1}{p_i} g_i^2 + \lambda \sum_{i=1}^{N} \frac{1}{p_i} h_i^2 + \gamma \left( \sum_{i=1}^{N} p_i - N \cdot s \right) - \sum_{i=1}^{N} \tau_i p_i - \sum_{i=1}^{N} \eta_i (1 - p_i), \, \tau_i \geq 0, \, \eta_i \geq 0, \, \forall i \tag{2}$$

Necessary conditions for solution of 1 are set by Karush–Kuhn–Tucker conditions:

$$\begin{cases} \frac{\partial \mathcal{L}}{\partial p_i} = -\frac{g_i^2}{p_i^2} - \lambda \frac{h_i^2}{p_i^2} + \gamma - \tau_i + \eta_i = 0, \forall i \\ \tau_i p_i = 0, \forall i \\ \eta_i (1 - p_i) = 0, \forall i \end{cases} \tag{3}$$

*The study was done while working at Yandex

Analyzing these conditions, it is easy to conclude that optimal solution has the following properties.

1. Since every $p_i > 0$, $\tau_i = 0$, $\forall i$.

2. If $\eta_i > 0$, then $p_i = 1$ and $g_i^2 + \lambda h_i^2 = \gamma + \eta_i > \gamma$.

3. If $\eta_i = 0$, then $p_i = \frac{\sqrt{g_i^2 + \lambda h_i^2}}{\sqrt{\gamma}} \leq 1$

Putting all together, there exists a threshold $\sqrt{\gamma}$, which divides sample into two parts: $\{x_i : \sqrt{g_i^2 + \lambda h_i^2} > \sqrt{\gamma}\}$ of size $k(\gamma)$ with $p_i = 1$ and $\{x_i : \sqrt{g_i^2 + \lambda h_i^2} \leq \sqrt{\gamma}\}$ of size $N - k(\gamma)$ with $p_i = \frac{\sqrt{g_i^2 + \lambda h_i^2}}{\sqrt{\gamma}}$.

Therefore, it is sufficient to find $\gamma = \gamma^*$, such that $\sum_{i=1}^{N} p_i = N \cdot s$. Desired value of $\gamma^*$ can be found as a solution of:

$$\sum_{i=1}^{N} p_i = \sum_{i=1}^{N-k(\gamma)} \sqrt{\frac{g_i^2 + \lambda h_i^2}{\gamma}} + k(\gamma) = N \cdot s \qquad (4)$$

Existence and uniqueness of solution for $s \in [0, 1]$ follows from the monotonous decrease of the left side of equation as a function of $\gamma$.

Setting $\mu = \sqrt{\gamma^*}$ finishes the proof. $\qquad\square$

## 3   Experiments

We use grid search with 5-fold cross validation to find the best sampling parameters for each algorithm and sampling ratio. For MVS it is a logspace grid on $[10^{-6}, 10^3]$ for $\lambda$ parameter and $\{5 : 1, 4 : 1, 2 : 1, 1 : 1, 1 : 2, 1 : 4, 1 : 5\}$ for large and small gradients ratio for GOSS. For other parameters we use tuned parameters from the publicly available benchmarks [1].

For the most visible demonstration of the superiority of MVS we place here charts of quality on sampling ratio dependence for every dataset from the main paper.

amazon

click

internet

kddchurn

kick

upsel

## References

[1] Catboost. 2018. Data preprocessing. `https://github.com/catboost/catboost/tree/master/catboost/benchmarks/quality_benchmarks`. (2018).