[Reviews · NeurIPS 2019]

Reviewer 1



To build gradient boosted decision trees, best possible feature split should be determined with the cost proportional to the number of examples. Sub-sampling from the training data can be carried out to make this task applicable for big datasets and improve the generalization ability of the model. This paper presents a non-uniform sampling strategy for stochastic gradient boosting (SGB). Namely, probabilities which are used for weighted sampling from the original training data are optimized with the objective of maximizing the estimation accuracy of splitting scores in the trees. Also, a closed-form solution is presented for this optimization problem. However, in the solution, the threshold is a parameter that should be tuned, where authors suggested using binary search. In this algorithm, the examples with gradient lower than a threshold are selected with probability equal to one, and examples with gradients greater than the threshold are selected using the probabilities obtained from the solution of the presented optimization formulation. The issue addressed in this paper is interesting and optimizing the sampling probabilities in GBDT is understudied as it is also mentioned in the paper. In my opinion, the author can re-organize the paper to make it more clear. Namely, pseudo-code of the proposed algorithm can be included in the text. Additionally, in the experiments section, there is a lack of explanation and discussion of the results (e.g. Figure 1 doesn't have any explanation). The lack of space can be compensated by eliminating Gradient gradient boosting pseudo-code. Also, the proof for Theorem 1 can move to the supplementary materials. In the learning time comparison, the tuning time is not included (or mentioned). In MVS, the sampling ratio is tuned, which means that we are fitting a smaller portion of data and consequently training will be faster. At least, this issue should be mentioned in the explanation.

Reviewer 2



Update: I read authors' responce RE:sampling rate does not tell the whole story - i was suggesting to add information about on average how many instances were used for each of the splits (because it is not equal to sampling rate * total dataset size). I am keeping my accept rating, hoping that authors do make the changes to improve the derivations/clarity in the final submission Summary: this paper is concerned with a common trick that a lot of GBDT implementation apply - subsampling instances in order to speed up calculations for finding the best split. The authors formulate the problem of choosing the instances to sample as an optimization problem and derive a modified sampling scheme that is aimed at mimicking the gain that would be assigned to a split on all the of the data by using a gain calculated only on a subsampled instances. The experiments demonstrate good results. The paper is well written and easy to follow, apart from a couple of places in derivations(see my questions). The method is more theoretically grounded than the common heuristics used and seems effective (it is compared with GOSS which was shown previously to do better than a commonly used trick to calculate the quantiles and use them as splits). Detailed comments 1) The formulation you are dealing with fits a learner to mimmic a gradient. However most of the implementations use a second order taylor decomposition of the loss (xgboost, lightgbm, tfbt). I assume you can incorporate the hessian the same way - your gain would be sum of squared gradients / sum of hessians. Any reasons why you didn't do that? 2) Line 24 i find this statement somewhat inaccurate. Most implementations don't consider every possible value for each split. Instead, a fixed number of quantiles (say 100) are precomputed and used for split evaluation. you can consider quantile calculation another form of sampling of instances (at least for the part that determines the split). This is not uniform sampling (it is based on frequency). Further, XGBoost for example  when calculating quantiles, assign weights to instances based on their hessian, so it even more complicated 3) Derivations: Line 164 formula 4 - i am not sure how u go from 7 to 8. c_l can be either negative or positive, so when it is negative, the upper bound should be even larger than in 7 Line 167 replacing values on the leaves cl with a constant - i am completely missing an intuition why you think it is valuable. The leaf value depends on the gradients of the instances that fall into that leaf, and moreover, on the mix of gradients. You can replace the counts of instances that fall into a leaf with a constant that depends on the sampling rate, but not the actual values of the leaves 4) Experiments: - in Table 3 or4 you should also report complexities of methods and how many actual instances were sampled (sampling rate does not tell the whole story) - I assume you don't include as a baseline the splits off the quantiles because GOSS was shown to perform better? I think it should be mentioned, because GOSS is actually not commonly used and quantiles are still the dominant way of choosing splits. Alternatively you can include the comparison with the quantiles calculated when instances where waited by their gradient, to make it more comparable to what you do) - You need to report SE in Table 3 Minor: line 131 which overcome ->which overcomes line 151 leave->leaf line 260 capable for -> capable of

Reviewer 3



originality: Novel. The idea of Minimal Variance Sampling is interesting. quality: Good. Both theoretical analysis and explicit benchmark show the proposed MVS can work. clarity: This paper is well written and organized overall. significance: somehow significantly. The experiments show that MVS use few samples for the training of GBDT, and thus is faster. However, the overall speed improvement seems not very significant. cons: 1.The idea of MVS is good, that is the sampled data should be able to approximate the expected squared deviation best. However, the derivation of MVS is not totally rigorous. In Theorem 1, it seems that the authors assume the tree structures are the same with subsampled and full training data. However, training a tree with full and subsampled data can result in different tree structures. In this sense, the \delta defined in the paper is not a true deviation. Assuming that the tree structure is not changed due to subsampling is also acceptable though, otherwise, the analysis could be complex. But it would be better to claim this assumption in the paper. 2. The datasets in the experiment part is relatively small. It would be more interesting to see how MVS affects the training time of large datasets. 3. The source code is not submitted, neither an anonymous github link is provided. questions: 1. Line 175 to 186 is a little difficult for me to understand. In line 175, it is said that the first term of equation (9) is responsible for “gradient distribution over the leaves of the decision tree”. However, in (9) there’s no “leaves” being involved, everything is a summation over the training data. Line 178 claims that \lambda is a “trade-off between the variance of single model and the variance of ensemble”. Everything in the derivation of (9) only concerns the information of current tree to be trained, and has nothing to do with the whole ensemble of boosting (which consist of all trees). 2. There’s a typo in formula (8) and Line 168, it should be Var(y_L) instead of Var(y_1). In Appendix, formula (2) should have no “\gamma \ge 0” constraint, since the constraint in (1) corresponding to \gamma is an equality constraint.

[Author Response · NeurIPS 2019]

First of all, we want to thank every reviewer for valuable notes and comments. We will do our best to accommodate all of them to reorganize the paper, make it more clear and thoughtful.

**To Reviewer 1.**

In particular, we will discuss tuning time of the algorithms. Since GOSS and MVS have one additional hyperparameter (large gradients size for GOSS and $\lambda$ for MVS), the tuning times for them are approximately the same. We also found a parameter-free formula for $\lambda$ (see the details in the answer to Reviewer 2).

**To Reviewer 2.**

– *I assume you can incorporate the hessian the same way... Any reasons why you didn't do that?*

Our paper is based on a standard GBDT score function (as, e.g., in [21]). Recently, we tested our method on the score function with hessians. The algorithm is easy to derive from our paper, when you replace a leaf size in Eq. 6 with sum of hessians in the leaf. The solution remains the same except for replacing $\sqrt{g_i^2 + \lambda}$ with $\sqrt{g_i^2 + \lambda h_i^2}$ in Theorem 2. Performance of this hessian-based sampling is even better (see Table 1), and we will add these results to the paper.

– *replacing values on the leaves $c_l$ with a constant ... The leaf value depends on the gradients of the instances that fall into that leaf ...*

Since the structure of the tree is not known in advance we propose replacing $c_l^2$ with some constant upper bound. Recently, we also found a parameter-free formula for $\lambda$, which achieves near-optimal results (and even better than ones obtained via cross-validation, see Table 1, MVS Adaptive): we approximate $c_l^2$ by squared mean of all gradient's absolute values. We will add this to the paper.

– *Line 24 ... Most implementations don't consider every possible value for each split.*

We will fix this misleading statement, sorry.

– *i am not sure how u go from 7 to 8. $c_l$ can be either negative or positive...*

The middle term (the covariance) in (7) is bounded by the sum of variances (Line 163). Since the variance is not negative the inequality is true for arbitrary $c_l$.

– *in Table 3 or 4 you should also report complexities of methods and how many actual instances were sampled...*

The algorithms differ only in their sampling stages. As it mentioned in section 4.3, all algorithms have $O(n)$ complexity here, so the complexities of all algorithms with same sampling ratio are the same. Unfortunately, we are not sure what do you mean by "sampling rate does not tell the whole story". Please, specify this issue, we will try to fix it.

**To Reviewer 4.**

– *It would be more interesting to see how MVS affects the training time of large datasets.*

We have experiments with large datasets and we will necessarily add them to the paper. The results are consistent with statements from paper, e.g. on Higgs dataset we have -15% learning time for MVS and -10% learning time for GOSS.

– *it seems that the authors assume the tree structures are the same with subsampled and full training data...*

Yes, paper derivations are conditioning on fixed previous splits of the tree. We will add this assumption.

– *it would be better if the MVS could be merged into LightGBM repo, I am looking forward to using it.*

The source code is available as a fork of LightGBM repo on github (see line 241). The github link is anonymized, and we think it will be possible to merge the code after review period ends.

– *Line 175 to 186 is a little difficult for me to understand...*

The idea is this: if only the first addend is optimized, then trees from close iterations will be trained on approximately the same instances (gradients do not change much), so this leads to higher correlation between them and as a result to higher variance of the ensemble. The second addend is minimized when all probabilities are the same.

| Sample rate | 0.02 | 0.05 | 0.1 | 0.15 | 0.2 | 0.25 | 0.3 | 0.35 | 0.4 | 0.5 |
|---|---|---|---|---|---|---|---|---|---|---|
| MVS | +13.96% | +8.21% | +4.60% | +2.39% | +1.23% | +0.38% | +0.00% | -0.17% | -0.24% | -0.44% |
| MVS adaptive | +13.89% | +7.57% | +3.87% | +1.72% | +0.64% | +0.16% | -0.06% | -0.19% | -0.29% | -0.50% |
| MVS with hessians | **+13.72%** | **+7.47%** | **+3.71%** | **+1.70%** | **+0.55%** | **-0.03%** | **-0.07 %** | **-0.28%** | **-0.32%** | **-0.51%** |

Table 1: Relative error change, average over datasets

[Meta-Review · NeurIPS 2019]

The authors propose a non-uniform sampling strategy for stochastic gradient boosted decision trees. In particular, sampling probability of the training data is optimized towards maximizing the estimation accuracy of the splitting score of decision trees. The optimization problem allows an approximate closed-form solution. Experiment results demonstrate superior performance of the proposed strategy. The reviewers agree that the paper can not only help understand sampling within GBDT from a more rigorous perspective but also improve GBDT implementations in practice. The reviewers have concerns about the structural assumption (between tree with sub-sampled data and tree with full data), the clarity of writing in some parts and the tuning time of the strategy. The authors are encouraged to improve on those parts, and contribute the implementation to open-source packages.